# Two- and Three-Dimensional Benchmarks for Particle Detection from an Industrial Rotary Kiln Combustion Chamber Based on Light-Field-Camera Recording

**Markus Vogelbacher** [1,*] , **Miao Zhang** [1] , **Krasimir Aleksandrov** [2] , **Hans-Joachim Gehrmann** [2] **and Jörg Matthes** [1]

[1] Institute for Automation and Applied Informatics, Karlsruhe Institute of Technology (KIT), Hermann-von-Helmholtz-Platz 1, 76344 Eggenstein-Leopoldshafen, Germany

[2] Institute for Technical Chemistry, Karlsruhe Institute of Technology (KIT), Hermann-von-Helmholtz-Platz 1, 76344 Eggenstein-Leopoldshafen, Germany

* Correspondence: markus.vogelbacher@kit.edu; Tel.: +49-721-608-25750

**Abstract:** This paper describes a benchmark dataset for the detection of fuel particles in 2D and 3D image data in a rotary kiln combustion chamber. The specific challenges of detecting the small particles under demanding environmental conditions allows for the performance of existing and new particle detection techniques to be evaluated. The data set includes a classification of burning and non-burning particles, which can be in the air but also on the rotary kiln wall. The light-field camera used for data generation offers the potential to develop and objectively evaluate new advanced particle detection methods due to the additional 3D information. Besides explanations of the data set and the contained ground truth, an evaluation procedure of the particle detection based on the ground truth and results for an own particle detection procedure for the data set are presented.

**Keywords:** benchmark; 3D point cloud; particle detection; light-field camera; refuse-derived fuels





## 1. Introduction

Image processing and computer vision are well-researched areas in which a variety of methods are available in a wide range of applications, e.g., engineering, medicine, and biology. However, as the developments in the field of machine learning in recent years have shown, new methods continue to be developed on a regular basis. In order to be able to evaluate the general performance of the new methods, data sets are needed that pose different challenges to the methods. For evaluation, the origin of the data, i.e., the application from which the data was generated, plays a rather minor role. In this article we present such a data set for methods in the field of detection of small objects/particles in 2D image data but also 3D point clouds.

We consider the detection of small fuel particles in a special combustion chamber, the rotary kiln, which is used industrially, e.g., for cement production or hazardous waste incineration but is also widely used for research purposes. Particle detection is used, for example, to determine the trajectory or the landing point of the fuel and thus to characterize the fuel and contribute to an optimization of the combustion process in comparison with CFD simulations [1,2]. Due to the difficult environmental conditions at the combustion chamber, the 2D and 3D data acquisition is not performed with a stereo camera system, which requires two access points to the combustion chamber, but with a light field camera.

At first, the detection of small particles has been mainly considered in the literature in the analysis of microscope images. Particle-like objects in this context represent complete

cells, cell nuclei, or other molecular particles in cells, depending on the magnification. An overview of different methods for particle detection in microscope images is given in [3] and specifically for cell nucleus detection in [4]. After background subtraction, single cells are detected via a gray level threshold in [5] and different types of particles are detected in microscope images in [6]. Ref. [7] uses Laplacian-of-Gaussian filters for the detection of cell nuclei. A tool for bacterial cell detection and subsequent analysis is presented in [8]. Machine learning methods are used in [9] to detect molecular particles in cells. Ref. [10] gives an overview on the use of deep learning for detection in microscope images, among others. Initial studies with fuel particles were performed in [11] for the trajectory of burning particles. Based on high-speed images, fuel particles in an afterburner chamber were considered in [12] and in a drop shaft in [13]. In all studies, the detection of the particles and their trajectories is performed using a gray-scale threshold. Refs. [11,12] also use background subtraction as image preprocessing. The variety of methods that can be used to detect small objects is shown by the overviews of current methods in [14,15]. The presented methods mainly make use of deep learning based approaches. Fields of application are, e.g., also aerial images [16]. In the field of particle detection in 3D point clouds, clustering methods are widely used. Clustering methods as described in [17–19] are applicable for different tasks.

Previously published benchmark datasets for particle detection also include mainly microscope images in the bioimaging area. Ref. [20] presents a benchmark dataset for biological image segmentation, including images with detected cell nuclei. The Particle Tracking Challenge data [21] can also be used as a benchmark for particle detection. This contains real image data of moving viruses, vesicles, receptors, and microtubule tips. Benchmark datasets for evaluating clustering methods are presented in [22–24], these are mostly based on synthetically generated data. Benchmark data sets with fuel particles and with a combination of 2D and 3D data for the detection of small objects are not yet available.

The new benchmark data set in a combustion chamber environment presented in this paper offers specific challenges that complicate the use of existing particle detection techniques. First, the design of the combustion chamber as a rotary kiln results in a constant variation of the background due to the rotary motion. Furthermore, the images contain a burner flame with similar gray values as the burning fuel particles. In addition, the fuel particles being detected are only a few pixels in size and do not have significant particle properties, such as shape or texture. Specific challenges of the dataset are also that the fuel particles are detectable as both burning and non-burning in the images, and thus sometimes appear lighter or darker compared to the image background. Furthermore, a distinction should be made between particles in the air and on the rotary kiln wall. Another special feature of the data set is the recording technique of the scene with a light-field camera, which provides 2D data in the form of a gray scale image as well as the 3D point cloud. It should be noted, that in 3D it is precisely the depth information of the light-field camera, i.e., the coordinate along the camera's line of sight, that is subject to strong fluctuations. The benchmark data set thus offers the possibility to test particle detection methods in 2D and 3D on the basis of challenging data and, in addition, to develop and quantitatively evaluate new methods due to the 3D information not yet available in other data sets. However, with the help of the benchmark data set, no statement about the general condition of the combustion process can be derived.

In the following, we first explain the creation of the benchmark data set in Section 2. This includes the description of the experimental setup of the test facility, the camera technology, and a first presentation of the data to facilitate understanding about the origin of the data and image content. Section 3 shows the contents of the benchmark dataset and thus, in addition to the image data, a description of the ground truth provided. Section 4 presents an evaluation procedure using the ground truth and Section 5 presents proprietary particle detection methods and results for the dataset. A summary is given in Section 6.

## 2. Experimental Setup for Data Generation

The benchmark data set for particle detection was developed in the context of experiments with refuse-derived fuels (RDF) at the 2 MW rotary kiln test facility BRENDA at the Institute of Technical Chemistry of the Karlsruhe Institute of Technology (KIT). Figure 1 shows the setup of the part of the plant relevant for the experiments, consisting of a rotary kiln with 8 m length and an inner diameter of 1.4 m and the subsequent afterburning combustion chamber.

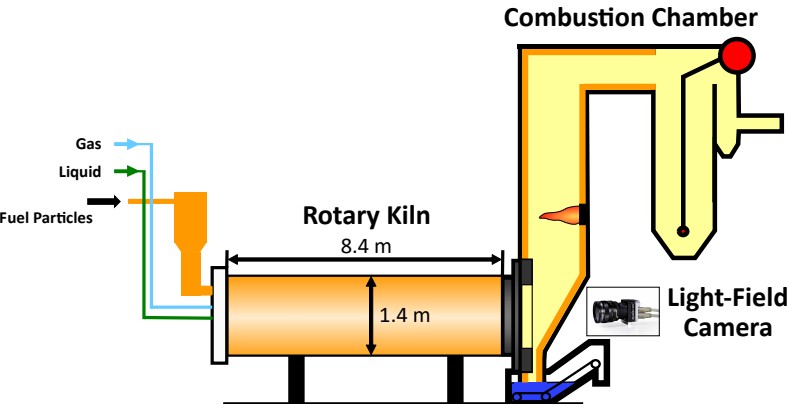

**Figure 1.** Construction of the BRENDA test facility at the Institute of Technical Chemistry of the Karlsruhe Institute of Technology (KIT).

At a temperature of about 1000 °C of the inner wall of the rotary kiln, RDF particles are injected into the rotary kiln via a lance at conveying air pressures of 4 bar to 5 bar on the inlet side of the rotary kiln. The *FLUFF* is used as RDF, which consists of RDF fractions capable of flight (including plastics or industrial and commercial wastes). Examples of the fractions that may be contained in the *FLUFF* fuel in various proportions are given in Figure 2. Figure 3 shows an example of the real *FLUFF* fuel mixture composed of the fractions as used in the experiments.

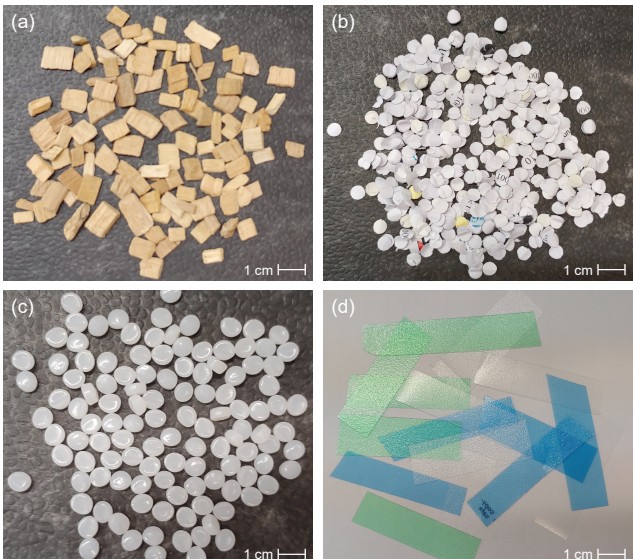

**Figure 2.** Examples of fuel fractions contained in the *FLUFF*. (**a**) Wood chips. (**b**) Paper. (**c**) PE granules. (**d**) Plastic films.

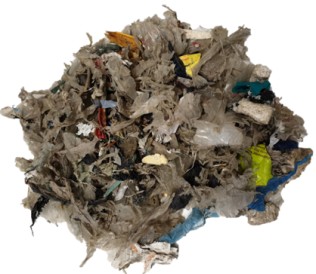

**Figure 3.** Example of the *FLUFF* fuel mixture [25].

When injected into the hot rotary kiln environment, the heterogeneous fuel mixture *FLUFF* causes particles of different sizes with different ignition behaviors to become visible. Figure 4 schematically represents the experimental sequence for a single RDF particle, first showing a non-burning particle in the air, which ignites in the air, then lands on the inner wall of the rotary kiln and continues to burn or ignite there. Furthermore, included in the schematic representation are the rotary kiln coordinates: lateral deviation x, height y, and depth z.

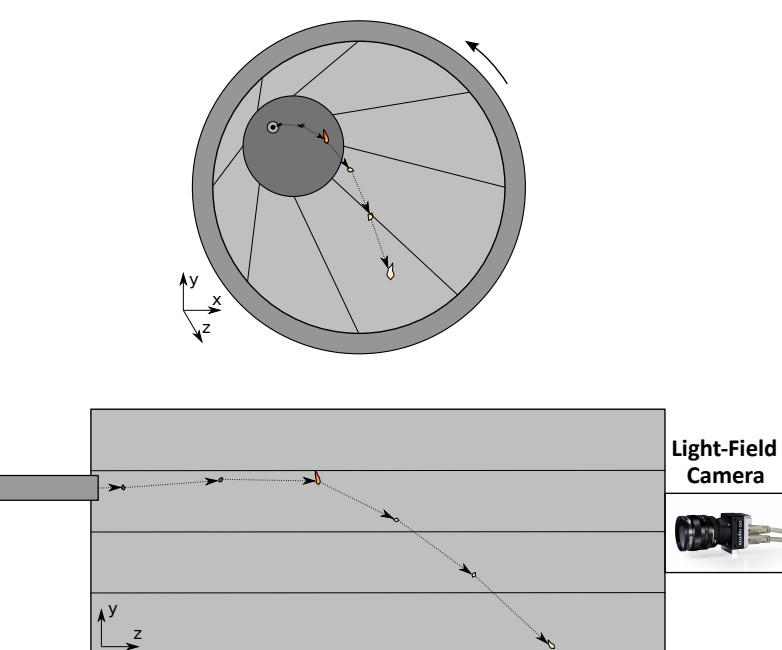

**Figure 4.** Schematic representation of one small fuel particle in a rotary kiln.

The flight trajectory of the particles can be observed via a camera at the outlet of the rotary kiln with the beginning of the afterburning combustion chamber, as shown in Figures 1 and 4. Due to structural limitations at a rotary kiln, the use of a stereo camera system, which requires two visual access points into the rotary kiln, to obtain 3D information of the scene is not possible. For this reason, a light-field camera, also called a plenoptic camera, is used. This makes it possible to observe the scene from different angles via a microlens array in front of the image sensor and thus derive 3D information via one visual access. By using microlenses with different focal lengths (multi-focus plenoptic camera), both a large depth of field range and a high maximum lateral resolution are achieved [26]. Metric depth information can also be obtained via calibration [27]. The used light-field camera R12 from Raytrix has a frame rate of 330 frames per second, a resolution of 2048 × 1536 pixels and microlenses with three different focal lengths. Thus, the light-field camera provides both 2D information through a gray scale image in image coordinates (u, v) (Figure 5a), as with a conventional camera in the visual field of view,

and 3D information through a 3D point cloud in the rotary kiln coordinate system (x, y, z) (Figure 5b).

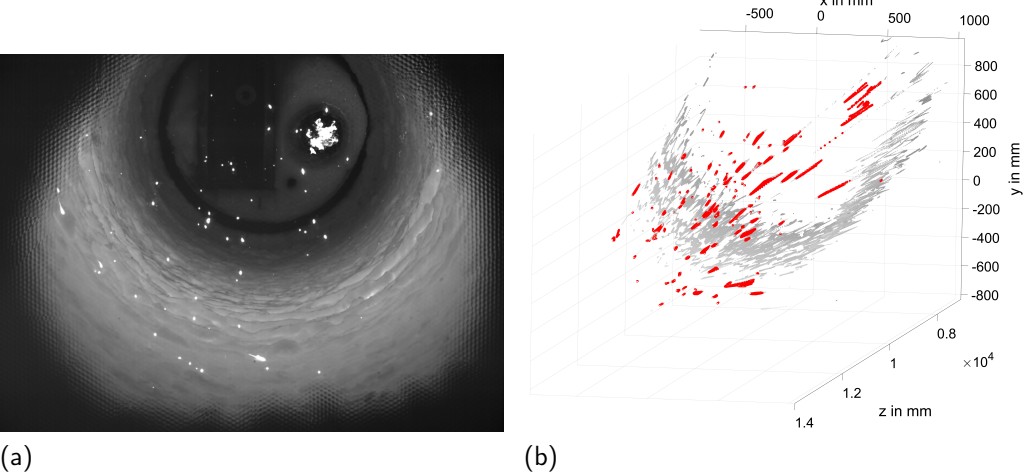

(a)           (b)

**Figure 5.** Light-field camera data in a hot rotary kiln environment: (**a**) 2D gray scale image and (**b**) 3D point cloud (RDF particles were manually marked in red and the rotary kiln in gray for better visualization).

The gray scale image shows the rotary kiln environment, the burner flame and the RDF particles. The 3D point cloud is given as a 3-channel image with the same image size as the gray scale image. The 3 image channels each contain the x, y, and z position in millimeters in 3D space for the image column u and image row v, providing a conversion from image to rotary kiln coordinates. However, 3D information is not available for all points in the 2D image. For the considered experimental scenario, 3D information is available for about 3% to 12% of the pixels. The coverage with 3D information can also be traced using a depth map. Figure 6 shows a depth map derived from the 3D data in false color representation, which contains in each pixel, if available, the corresponding depth coordinate from the 3D point cloud. Dark blue areas mark image pixels for which no depth and therefore no 3D information is available. It should be noted that due to the camera-internal position of the coordinate origin, the outlet point of the lance, via which the particles are blown into the rotary kiln, is approximately at a depth of 6 m.

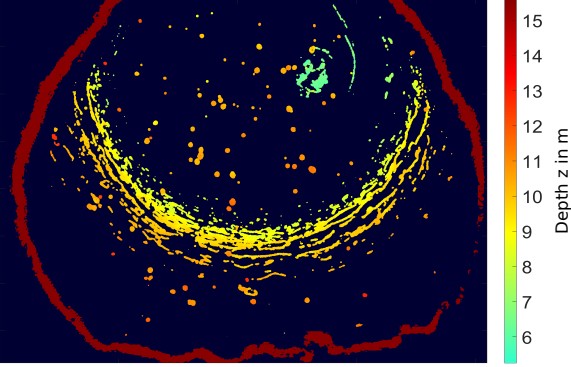

**Figure 6.** Depth map in false color display for checking the 3D information of the light-field camera.

The partial coverage with 3D information is due to the fact that 3D information can only be determined in areas with sufficient structure or at edges. For homogeneous image areas and in the border area of the image, less or inaccurate 3D information is available. Therefore, reducing the image area to a suitable region of interest (ROI) when using the 3D information is recommended and is presented below in Section 3.2. Image areas with

particles contain to approx. 92% and thus significantly more often a 3D information (see Section 3).

### 3. Benchmark Data Set

From the previously described experiments a sequence with 2010 images was recorded for particle detection algorithms. From these images five sequences with 10 images each (total 50 images) were selected and an associated ground truth was manually determined, containing a total of 5701 particles across all images. Due to the frame rate of the camera, the images of a sequence have a time interval of about 3 milliseconds. In the benchmark dataset, the complete raw data of the camera for all sequences are available, i.e., both gray scale images and 3D point clouds (Figure 5). An overview of the provided data for the benchmark are listed in Table 1. In the following, the ground truth is explained in more detail.

**Table 1.** Overview of benchmark data set.

| Data | Format | Size |
|---|---|---|
| Gray scale image | *PNG* | 2048 × 1536 pixel |
| Ground Truth classification (Labels) | *PNG* | 2048 × 1536 pixel |
| List of Particle Position | *TXT* | |
| 3D point cloud | *TIFF* | 2048 × 1536 × 3 pixel |

### 3.1. Ground Truth

For the use case of flying RDF particles in a rotary kiln environment, the classes listed in Table 2 result for each pixel.

**Table 2.** Classes for flying RDF particles in a rotary kiln environment and their associated labels in ground truth.

| Class | Label |
|---|---|
| rotary kiln | 0 |
| burner flame | 1 |
| burning particle in air | 2 |
| non-burning particle in air | 3 |
| particle on wall | 4 |

The rotary kiln environment is considered as the background. The flame visible in the images contributes to a constant inner temperature of the rotary kiln during the experiments. For the detection of particles, this is considered as a disturbing signal, since it has similar gray values as burning particles and can thus cover particles. The class particle on wall contains only burning particles, since due to the high wall temperature of the rotary kiln RDF particles ignite and burn immediately when landing on the wall. RDF particles of interest for particle detection can therefore be divided into three classes, depending on whether they are burning or not and whether they are in the air or on the wall. Burning particles have a significantly higher gray value than non-burning particles and can therefore be classified by their gray value. A distinction between particles in the air or on the wall can be made, e.g., via the 3D information. After a suitable detection of the rotary kiln geometry in the 3D point cloud, an appropriate classification can be performed based on the distance to the rotary kiln. Alternatively, the motion of a particle could also be analyzed via the image sequence. Particles in the air move much faster than particles on the rotary kiln wall.

For each gray scale image, a labeled ground truth image is created in which each pixel is assigned to a class. The labeled image has the same image size of 2048 × 1536 pixels as the original image and the classes are defined by the gray value of the pixels. The gray values for each class correspond to the label number in Table 2. Figure 7 shows an

example of 2D ground truth as an overlay over the corresponding gray scale image from the raw data.

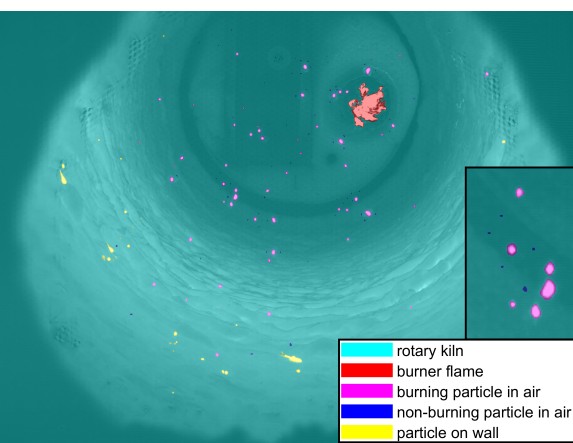

**Figure 7.** Example of the labeled ground truth image for the entire image and a cropped image area as an overlay over the corresponding gray scale image from the raw data.

In addition to the completely labeled image, the position of the particles contained in the image can also be specified using the coordinates of the center of gravity in image coordinates (u, v). These coordinates are available in separate *TXT* files as lists for burning particles in the air, non-burning particles in the air and particles on the wall. In addition to the coordinates, these lists also contain in a third column the information whether 3D information from the point cloud is available for the particle. If 3D information is available for the coordinate, a 1 is entered in the third column of the corresponding line. If no 3D information is available, a 0 is entered. Figure 8 shows an excerpt from one of the described *TXT* files for burning particles with the particle coordinates in the first two columns and the binary parameters for the 3D information in the third column.

```
Number of burning particles in the air: 58

Position of particles: row / pixel, column / pixel, 3D info

500     487     1
994     547     1
355     571     1
336     585     1
613     625     1
526     628     1
582     632     1
```

**Figure 8.** Extract from a *TXT* file of the particle detection. With the coordinates of the center of gravity of a particle in the first two columns and a binary parameter indicating whether 3D information is available for the coordinate in the third column.

Table 3 provides an overview of the number of particles contained in the ground truth. The number is given for the different particle classes with and without 3D information.

**Table 3.** Number of particles in ground truth of the benchmark data set.

| Particle Class | Total | With 3D Info | |
|---|---|---|---|
| burning in air | 2454 | 2439 | 99.39% |
| non-burning in air | 1466 | 1155 | 78.79% |
| on wall | 1781 | 1658 | 93.09% |
| in air | 3920 | 3594 | 91.68% |
| all classes | 5701 | 5252 | 92.12% |

### 3.2. Supplementary Notes on Data

Due to the difficult environmental conditions in a hot rotary kiln environment and the small burning bright particles, the 3D information of the light-field camera is subject to strong fluctuations especially in depth direction z. This becomes clear when viewing the 3D point cloud, as seen in the example in Figure 5b. Individual particles are not characterized by a tight cluster of points in all coordinate directions but form a long tail of points especially in the depth direction z. This appearance must be taken into account in detection methods that use 3D information.

Furthermore, we would like to point out once again that the three-channel image of the 3D point cloud can be used as a look-up table for the conversion of the 2D image coordinates (u, v) into 3D rotary kiln coordinates (x, y, z). For each image point (u, v), the corresponding (x, y, z) position can be read from the *TIFF* image of the point cloud at the same location using the three image channels. However, especially in the border area of the image, i.e., at the transitions to the black image area, no or incorrect (artifacts in the image) 3D information is available. These image areas should not be considered for further evaluation that uses the 3D information. For this reason, a suitable ROI is defined using a polygon course (Figure 9, yellow). This ROI is supplemented by a circular disturbance region of the burner flame (Figure 9, red). The information about this ROI is included in the benchmark data set.

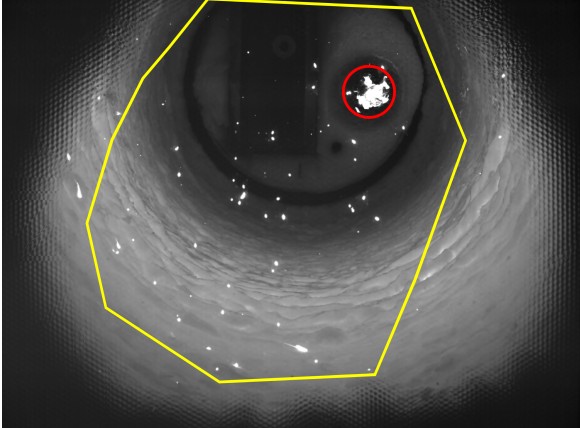

**Figure 9.** Region of Interest for particle detection.

In the image area of the ROI, 3D information is available for an average of 12% to 28% of the pixels. Compared to the complete image, the availability of 3D information for image regions with particles can also be increased to almost 95% by using the ROI (Table 4). The use of ROI ensures comparability between newly developed detection methods based on the benchmark dataset. However, the number of particles included in the ground truth is reduced by using the ROI (Table 4).

**Table 4.** Number of particles in ground truth of the benchmark dataset when using region of interest.

| Particle Class | Total | With 3D Info | |
| --- | --- | --- | --- |
| burning in air | 1698 | 1690 | 99.53% |
| non-burning in air | 585 | 480 | 82.05% |
| on wall | 1376 | 1304 | 94.77% |
| in air | 2283 | 2170 | 95.05% |
| all classes | 3659 | 3474 | 94.94% |

For methods that use only the 2D information from the gray scale image, it is not necessary to use the ROI defined by the polygon course. Ground truth data are also available for the complete image area outside this ROI.

### 3.3. Organization and Visualization

The presented benchmark data for particle detection can be obtained under [28]. Thereby the provided folder contains the following subfolders analogous to the data listed in Table 1:

01_Images;

02_Labels;

03_Particle_List;

04_Point_Cloud;

05_Matlab;

06_All_Data.

The 05_Matlab contains, besides the information about the proposed ROI as *TXT* file and mask, the Matlab visualization script for displaying the 2D ground truth and the ROI as used, for example, in this paper. Here, the centroids of the particles from the particle lists can also be drawn into the labeled image (Figure 10).

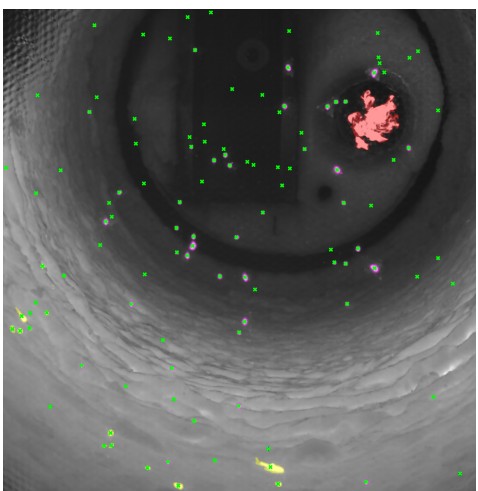

**Figure 10.** Result of the image display via the visualization script with the particle detections marked (green) and the labeled areas of the particles (purple, blue, yellow) as in Figure 7 (cropped image area).

A script is also provided for evaluating the results of a particle detection procedure in image coordinates using ground truth. The evaluation method used in this script is explained in more detail in the following section. The scripts provided were developed and tested using Matlab R2021a.

In 06_All_Data, the gray scale images and the 3D point clouds are given for the complete dataset (2010 images), i.e., also for the images without ground truth. Furthermore, included is a *TXT* file that describes the position of the ground truth data in the complete image sequence.

### 4. Evaluation Method for Benchmark

The evaluation method presented in this section is used to determine the performance of new particle detection methods. The use of this method should contribute to a better comparability of the detection methods.

The Matlab script Evaluation_Framework.m included in the benchmark dataset with the associated function PD_Evaluation.m allows the calculation of the performance of new detection methods by comparing the detection results with the lists of particle positions included in the benchmark. For the use of the Matlab script, it is intended to enter the particle positions obtained by a new detection into a similar list and load them accordingly

in the script. Figure 11 shows the data flow diagram of the evaluation method with the corresponding Matlab scripts.

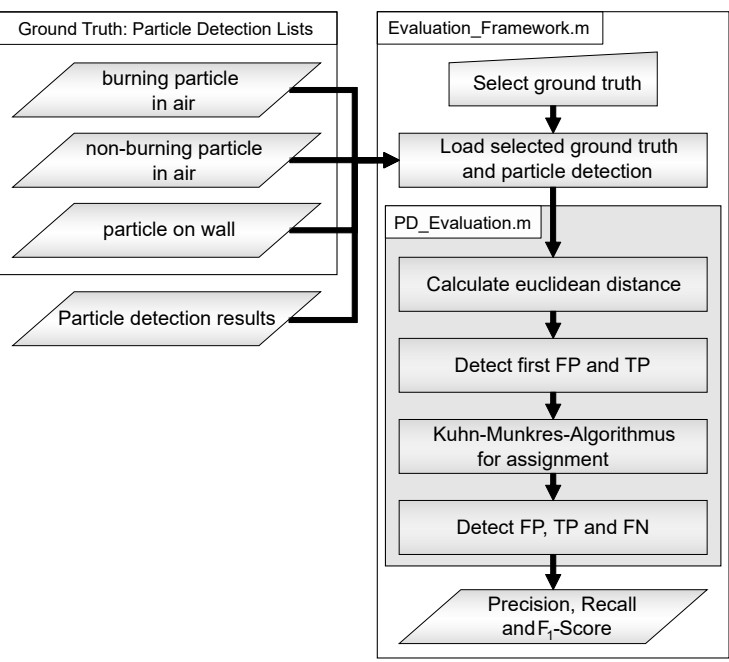

**Figure 11.** Data flow diagram for particle detection evaluation method.

Since different classes of particles are present and labeled in the benchmark data set, different tasks can be considered, such as the detection of all particles or the detection of only particles that are in the air. For this reason, the first step is to select and compile the desired ground truth. For example, to evaluate all particles in the air, the list for burning particles in the air and non-burning particles in the air must be merged. In the Matlab script provided, there are four setting options for this for burning particles in the air, non-burning particles in the air, all particles in the air, and all particles (in the air and on the wall). After selection of the desired particle Ground Truth all selected particles are merged. A distinction in the class of origin is then no longer provided. An evaluation for several classes is obtained by multiple execution of the evaluation method.

The most important step for the evaluation is the assignment of the particle positions from the detection and the ground truth. For this purpose, the Euclidean distance between all particles of the detection and the ground truth is determined. Then, each detection is assigned to all particles of the ground truth that have a Euclidean distance smaller than a previously defined threshold, e.g., 15 pixels. Thus, detections are obtained to which one particle, several particles or no particles from the ground truth could be assigned. Detections without an assignment from the ground truth can be detected as false and thus directly counted as false positives (FP). Detections that have been assigned to a single particle from the ground truth and this particle does not meet the distance condition for any other detection can be directly detected as correct and thus counted as true positives (TP). By preprocessing these first FP and TP detections, the complexity and thus the effort for further assignment can be reduced. The remaining detections that have several possible ground truth particles in their neighborhood or that share a ground truth particle are subjected to further investigation. Using the James Munkres variant of the Hungarian assignment method [29], also called the Kuhn–Munkres algorithm, an optimized assignment based on a global nearest neighbor approach can be performed using the Euclidean distances as costs. The cost matrix used for this optimization is composed of the Euclidean distances $d_{i,j}$ between the detections $P_i$ and the ground truth particles $G_j$. Figure 12 shows the structure of the cost matrix.

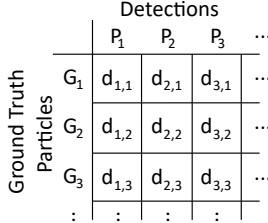

**Figure 12.** Set up the cost matrix for the Kuhn–Munkres algorithm.

If a detection cannot be assigned to a ground truth particle, the previously defined threshold value of the maximum Euclidean distance is entered as cost $d_{i,j}$ at the corresponding position and thus penalized. Thus, as many detections as possible are assigned to the ground truth. Detections that have been assigned to a ground truth particle via the algorithm are added as TP and detections that have not been assigned to a particle are added as FP. The number of non-detected ground truth particles and thus the false negatives (FN) result from the difference between the total number of particles contained in the ground truth and the number of TP from the evaluation.

Using the results for TP, FP, and FN for all benchmark data, the parameters Precision, Recall, and $F_1$-Score [30], which is the harmonic mean of Precision and Recall, are calculated to evaluate the performance of the detection method used via

$$\text{Precision} = \frac{\sum\limits_{i=1}^{N} \text{TP}_i}{\sum\limits_{i=1}^{N} \text{TP}_i + \sum\limits_{i=1}^{N} \text{FP}_i}, \tag{1}$$

$$\text{Recall} = \frac{\sum\limits_{i=1}^{N} \text{TP}_i}{\sum\limits_{i=1}^{N} \text{TP}_i + \sum\limits_{i=1}^{N} \text{FN}_i} \text{ and} \tag{2}$$

$$F_1 = 2 \cdot \frac{\text{Precision} \cdot \text{Recall}}{\text{Precision} + \text{Recall}} \tag{3}$$

with $N$ the number of images used for evaluation.

## 5. Methods for Particle Detection on Benchmark

Based on the benchmark data set, first detection methods were developed and tested in the course of creation [31]. Particles are detected both with 2D or 3D information only but also via a combination of 2D and 3D information.

### 5.1. Two Dimensional Particle Detection

The flow diagram of RDF particle detection for the 2D data from the benchmark dataset after [31] is given in Figure 13.

The brick lining and the buildup on the inner wall of the rotary kiln lead to structures in the camera image, which can cause false detections during particle detection. Since the particles have a faster motion compared to the rotary kiln, the rotary kiln area can be removed as an image pre-processing step by background subtraction, thus simplifying particle detection. It is important to note that the slow rotation of the rotating tube (approx. 0.2 rpm) also causes a variation of the background. Thus, only images close in time to the image under consideration can be considered for background formation. The background image (median) can be obtained, for example, by calculating the temporal median value for the gray level of each pixel over a sequence of images (for the evaluation in this paper 75 images before and after the considered image were chosen) and subtracting it from the

original image. An even smaller time range can be obtained by subtracting two ($I_{i-1}$ and $I_i$) or three ($I_{i-1}$, $I_i$ and $I_{i+1}$) consecutive images:

$$I_{\text{diff,2Img}} = I_i - I_{i-1} \text{ or} \tag{4}$$

$$I_{\text{diff,3Img}} = (I_i - I_{i+1}) + (I_i - I_{i-1}). \tag{5}$$

Due to the slow movement of the background, this method also removes particles that lie on the rotary kiln wall. After background subtraction, scale invariant feature transform (SIFT) [32] keypoints are determined for the image within the ROI. Due to the strong similarity of the Difference of Gaussian filter of the SIFT to the gray scale of a particle, the keypoints can be treated as particle detections for this use case. Since it can happen that, e.g., larger particles contain several keypoints, an appropriate post-processing of the keypoints is still accomplished on the basis of the gray values. Finally, the post-processing keypoints result in the position of the RDF particles.

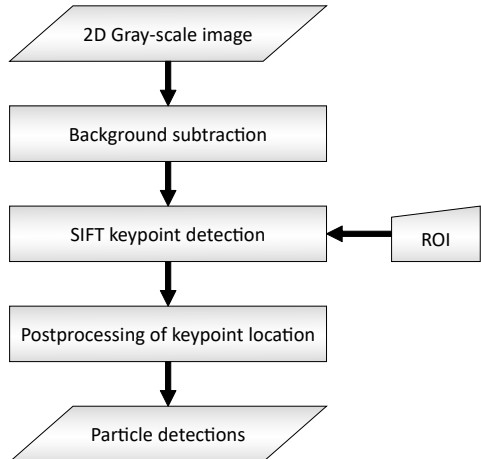

**Figure 13.** Flow diagram of the 2D particle detection.

*5.2. Three Dimensional Particle Detection*

Figure 14 describes the process of detecting RDF particles for the 3D data from the benchmark dataset according to [31].

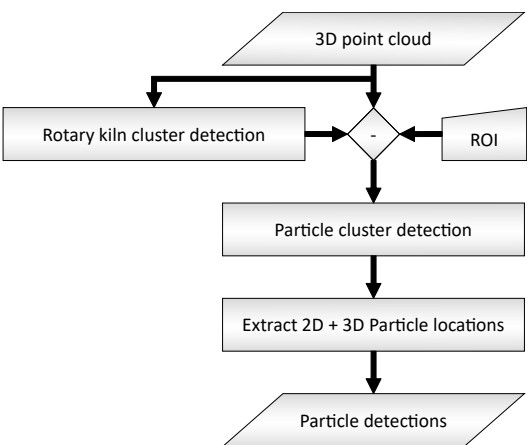

**Figure 14.** Flow diagram of the 3D particle detection.

Here, the DBSCAN (Density-Based Spatial Clustering of Applications with Noise) clustering algorithm [33] is used with different parameter settings. First, the largest cluster of the 3D point cloud, which can be considered as the inner wall of the rotary kiln, is detected using DBSCAN. The points of the rotary kiln cluster and points outside the ROI

can be removed from the 3D point cloud for particle detection. Particle clusters can then be detected via reapplying DBSCAN with parameter settings that favor particularly small clusters. The 3D positions of the centroids of the particles are returned and converted to 2D coordinates for comparison of the results.

### 5.3. Combination of 2D and 3D Particle Detection

Furthermore, in [31] it is described how the results from 2D SIFT and 3D DBSCAN particle detection can be combined to increase the performance of the detection. When combined, this can result in five different cases for the assignment of particles from each method:

1. exactly one SIFT particle $\leftrightarrow$ exactly one Cluster;
2. multiple SIFT particles $\leftrightarrow$ one Cluster;
3. no SIFT particle $\leftrightarrow$ one Cluster;
4. one SIFT particle $\leftrightarrow$ no Cluster;
5. one SIFT particle $\leftrightarrow$ multiple Clusters.

Case 1 leads directly to a reliable particle detection. For case 2, it must be checked whether there are several particles that could not be separated during clustering or whether there is one large particle that contains several keypoints. This is conducted by analyzing the gray value gradient within the cluster. Depending on whether multiple intensity maxima can be detected or not, multiple particle detections are taken or only a single particle. For case 3 and 4, the gray value gradient in the detection environment is also taken into account. By comparison with a Gaussian distribution it is decided whether it is a particle or a false detection (e.g., by noise). Case 5 occurs rarely and concerns only large burning particles. The missing 3D information in homogeneous areas of large particles can lead to a large particle being split into several smaller particles. The assignment and post-processing of all particles from 2D and 3D detection ultimately represents the detection result of the combination method.

### 5.4. Results

Table 5 shows the results from [31], among others, for different combinations of the previously listed methods when using all particles from the selected ground truth test data. For the evaluation, images 11 to 50 of the benchmark are used as training data to set the parameters of the different methods, maximizing the $F_1$-Score for the training data. Images 1 to 10 of the benchmark are used as test data to calculate Precision, Recall, and $F_1$-Score. The method described in Section 4 is used as the evaluation method.

**Table 5.** Comparison of different particle detection methods for all ground truth particles.

| | Precision | Recall | $F_1$-Score |
|---|---|---|---|
| Clustering | 42.09% | 71.47% | 52.98% |
| SIFT-Median | 97.65% | 85.69% | 91.28% |
| SIFT-2Img | 93.79% | 85.88% | 89.66% |
| SIFT-3Img | 92.07% | 88.11% | 90.04% |
| Clustering + SIFT-Median | 94.48% | 89.02% | 91.67% |
| Clustering + SIFT-2Img | 95.06% | 86.86% | 90.78% |
| Clustering + SIFT-3Img | 94.43% | 87.39% | 90.77% |

It can be seen from the results that the background subtraction 2Img and 3Img remove particles on the wall, resulting in a smaller number of TP and a higher number of FN. For this reason, the Precision values of the methods with these background subtractions are below those with the background subtraction median. For the distinction between particles in the air and on the wall, a classification of the particles connected to the detection is necessary, e.g., based on the 3D information of the rotary kiln wall as in [31].

It is also noticeable that especially small non-burning and therefore dark particles are a big challenge and lead increasingly to FN. In this case, a compromise must be made in

the parameter setting for the 2D detection methods that use only the gray level information. This is because an increased detection of the dark particles leads to more false detections, i.e., FP. This problem does not arise with the clustering algorithm, since it works independently of the gray values. Therefore, a combination with clustering leads to a slight improvement of the results. Overall, however, none of the methods can achieve $F_1$-Scores above 92%. The combination of SIFT with median background subtraction and DBSCAN clustering from [31] thus represents the initial best method for particle detection for all particles in the benchmark dataset. These initial detection results thus provide the opportunity for comparison with newly developed detection methods.

## 6. Conclusions

This paper presents a new benchmark dataset for fuel particle detection in a rotary kiln and provides a comparison for image processing methods under real industrial environmental conditions. The applied light-field camera also offers a novel potential of using 2D and 3D information for particle detection.

In addition to the description of the creation of the data set with 5701 particles, it was shown that the special application field and the camera technology also provide special challenges. Small particles with different gray value ranges due to combustion, an inhomogeneous background due to the rotary kiln or also a fluctuating depth information due to the light-field camera are mentioned as examples. The presented benchmark dataset contains labeled images, which assigns each pixel of the gray scale images to a class such as rotary kiln, particle or flame, and lists, which contain the coordinates of the different particle classes. For a comparison with the ground truth of the benchmark dataset, an evaluation method based on a global nearest neighbor approach is explained and also provided as a Matlab script. In addition, the particle detection methods created with the dataset provide initial results for the benchmark dataset with an $F_1$-Score of almost 92%.

In the future, it is hoped that many known and new methods will be tested and developed on the new benchmark dataset. Increasing the performance of particle detection could thus also support new insights in the field of RDF combustion in a rotary kiln environment. Especially for tracking the trajectory of RDF to derive characteristics of the fuel, an accurate detection of the particles is essential. It is planned to successively expand the benchmark dataset available online with new data.

**Author Contributions:** Conceptualization, M.V. and J.M.; methodology, M.V. and M.Z.; software, M.V.; validation, M.V., M.Z., K.A., H.-J.G. and J.M.; formal analysis, M.V.; investigation, M.V., M.Z. and J.M.; resources, M.V., K.A. and H.-J.G.; data curation, M.V. and M.Z.; writing—original draft preparation, M.V.; writing—review and editing, M.Z. and J.M.; visualization, M.V.; supervision, J.M.; project administration, J.M.; funding acquisition, J.M. All authors have read and agreed to the published version of the manuscript.

**Funding:** This study is supported by AiF—German Federation of Industrial Research Associations (No. 20410N).

**Institutional Review Board Statement:** Not applicable.

**Informed Consent Statement:** Not applicable.

**Data Availability Statement:** The data used in this article are available at [28].

**Acknowledgments:** We acknowledge support by the KIT-Publication Fund of the Karlsruhe Institute of Technology.

**Conflicts of Interest:** The authors declare no conflict of interest.

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
