# Peer review of "Two- and Three-Dimensional Benchmarks for Particle Detection from an Industrial Rotary Kiln Combustion Chamber Based on Light-Field-Camera Recording"

_data, 2022_

Round 1

Reviewer 1 Report

Comments:

1. In chapter Introduction line 17-21 authors tryied to explain the meaning of the problem and how to resolve this problem. But, in my opinion this part of text should be changed, because is about "nothing".

2. In my opinion, the article has scientific potential, but I propose to develop more than one frame / photo separately.

Variants can be considered:

(a) few different frames in one image (as it was done),

(b) each frame separately at a set time interval.

For me it is important in the analysis because it shows which combustion particles show up first and which next.

The combustion process is a process over time. Thus, the image captured by the camera will be different every second of burning (the used light-field-camera R12 100 from Raytrix has a frame rate of 330 frames per second).

Reviewer 2 Report

The manuscript describes a benchmark dataset created by authors for detection of fuel elements in a rotary kiln combustion chamber. A light field camera was used to generate 2D and 3D images. A ground truth dataset was also created by marking corresponding pixels. The benchmark dataset can be used to evaluate new particle detection methods.

1) The motivation of this work is insufficiently described. Detection of particles in a rotary kiln combustion chamber using light field camera is a very narrow special task. What is the reason for creating the proposed dataset? Should the light field camera be used for process monitoring or a redular camera is sufficient? What type of combustion process control requires a particles detection algorithm? I would recommend to improve the description of motivation of the work.

2) Insufficient description of motivation is also reflected in the references:
25% of references refer to the works of the author. Moreover all recent works
(the last 5 years) are only author’s works.  It looks like nobody else is
interested in the task.

3) The fig. 6 looks incorrect for me. This is because the authors included dark blue color (which means “dark area for which no depth is available”) into the false color scale. So it looks like “dark area” corresponds to 5 meters and the red contour around corresponds to 15 meters. I would recommend to distinguish false colors associated with depth and special colors indicating no data. For example, the scale of false color can be in the range “green-yellow-red” (no dark blue here).

4) It is not explained why the constant “50” is used in formulae (1) and (2)
on page 10 to define the sum range. If this is the umber of images
then using “N” works better.

Round 2

Reviewer 2 Report

The text of manuscript is improved. My remarks are responsed.